# The Inhibitory Receptor GPR56 (*Adgrg1*) Is Specifically Expressed by Tissue-Resident Memory T Cells in Mice But Dispensable for Their Differentiation and Function In Vivo

**DOI:** 10.3390/cells10102675

**Published:** 2021-10-06

**Authors:** Cheng-Chih Hsiao, Natasja A. M. Kragten, Xianhua Piao, Jörg Hamann, Klaas P. J. M. van Gisbergen

**Affiliations:** 1Department of Experimental Immunology, Amsterdam Institute for Infection and Immunity, Amsterdam University Medical Centers, University of Amsterdam, 1105 AZ Amsterdam, The Netherlands; c.hsiao@amsterdamumc.nl; 2Department of Hematopoiesis, Sanquin Research and Landsteiner Laboratory, Amsterdam University Medical Centers, University of Amsterdam, 1066 CX Amsterdam, The Netherlands; n.kragten@sanquin.nl; 3Department of Pediatrics, Newborn Brain Research Institute, Weill Institute for Neuroscience, University of California, San Francisco, CA 94158, USA; xianhua.piao@ucsf.edu

**Keywords:** adhesion GPCR, GPR56, inhibitory receptor, tissue-resident memory T cells

## Abstract

Tissue-resident memory T (T_RM_) cells with potent antiviral and antibacterial functions protect the epithelial and mucosal surfaces of our bodies against infection with pathogens. The strong proinflammatory activities of T_RM_ cells suggest requirement for expression of inhibitory molecules to restrain these memory T cells under steady state conditions. We previously identified the adhesion G protein-coupled receptor GPR56 as an inhibitory receptor of human cytotoxic lymphocytes that regulates their cytotoxic effector functions. Here, we explored the expression pattern, expression regulation, and function of GPR56 on pathogen-specific CD8^+^ T cells using two infection models. We observed that GPR56 is expressed on T_RM_ cells during acute infection and is upregulated by the T_RM_ cell-inducing cytokine TGF-β and the T_RM_ cell-associated transcription factor Hobit. However, GPR56 appeared dispensable for CD8^+^ T-cell differentiation and function upon acute infection with LCMV as well as *Listeria monocytogenes*. Thus, T_RM_ cells specifically acquire the inhibitory receptor GPR56, but the impact of this receptor on T_RM_ cells after acute infection does not appear essential to regulate effector functions of T_RM_ cells.

## 1. Introduction

Cytotoxic CD8^+^ T cells protect the body against pathogenic viruses and intracellular bacteria through the targeted production of cytolytic enzymes and inflammatory cytokines to infected cells and other immune cells, respectively. Cytotoxic CD8^+^ T cells are activated upon recognition of peptide antigens derived from viruses or bacteria in the context of MHC class I molecules through their T-cell receptors. These CD8^+^ T cells are also equipped with a comprehensive repertoire of inhibitory receptors that maintain quiescence at steady state but still allow rapid activation upon pathogen encounter [1]. Given that cytotoxic CD8^+^ T cells are also important for the clearance of tumor cells, current immunotherapies aim to exploit these cells for the benefit of cancer patients. In particular, tuning the balance between inhibition and activation of CD8^+^ T cells through blockade of immune checkpoints, such as PD-1, appears to be a promising strategy for the treatment of cancer patients [2].

We and others previously identified GPR56 as a surrogate surface marker indicating cytotoxic capacity across human lymphocyte subsets, including CD8^+^ and CD4^+^ T cells as well as NK cells [3,4,5,6]. GPR56 is an adhesion G protein-coupled receptor (GPCR) with established roles in brain development, hematopoiesis, male fertility, muscle hypertrophy, and tumor growth and progression [7,8,9]. We obtained evidence that GPR56 negatively regulates effector functions, including production of inflammatory cytokines and cytolytic enzymes, degranulation, and target cell killing in NK cells [10]. Expression of GPR56 in NK cells is driven by Hobit (homolog of Blimp in T cells; *ZNF683*), which is expressed in circulating cytotoxic CD4^+^ and CD8^+^ T cells [11,12]. This suggests that Hobit may drive expression of GPR56 in T cells and that GPR56 limits the potentially harmful cytotoxic effects of granzyme B and proinflammatory cytokines under homeostatic conditions when activity of these cytotoxic T cells requires restraints.

Circulating memory CD8^+^ T cells can be separated into central memory T (T_CM_) and effector memory T (T_EM_) cells. Both of these populations continually circulate through the blood, but T_CM_ cells migrate to lymph nodes and spleen, whereas T_EM_ cells migrate to the peripheral tissues. Recently, a novel population of memory CD8^+^ T cells has been described that, in contrast to T_CM_ and T_EM_ cells, does not circulate, and these memory T cells are therefore named tissue-resident memory T (T_RM_) cells [13]. T_RM_ cells are found predominantly in epithelial and mucosal tissues, such as the lungs, the small intestine, and the skin, where they are ideally positioned as a first-line defense against invading pathogens. T_RM_ cells have an important sentinel function, as they are able to alarm surrounding parenchymal cells and recruit circulating effector cells, such as neutrophils and monocytes through the immediate release of proinflammatory IFN-γ upon encounter of pathogens [14,15]. The strategic location at pathogen entry sites and their capacity to immediately initiate pathogen clearance make T_RM_ cells important immune weapons for protection against reinfection [15,16,17]. However, the long-term persistence of these proinflammatory memory T cells in peripheral tissues during the steady state also appears to require control mechanisms to prevent inadvertent immune activation. Indeed, T_RM_ cells highly express inhibitory receptors such as PD-1, which protect the peripheral tissues against proinflammatory actions of T_RM_ cells under homeostatic conditions [18,19,20].

Here, we addressed the contribution of GPR56, which we have previously identified as an inhibitory receptor suppressing the proinflammatory activity of cytotoxic lymphocytes [10], to the regulation of pathogen-specific T_RM_ cells. Exploring experimental models of acute viral and bacterial infection, we found that murine T_RM_ cells, in contrast to other memory T cells, specifically upregulated expression of GPR56. The upregulation of GPR56 was driven by the T_RM_ cell-inducing cytokine TGF-β and partially depended on the transcription factor Hobit. However, analysis of GPR56 gene-deficient mice infected with lymphocytic choriomeningitis virus (LCMV) or *Listeria monocytogenes* did not reveal an essential role for GPR56 in CD8 T-cell differentiation and function during the effector and memory phase.

## 2. Material and Methods

### 2.1. Mice

*Adgrg1*^flox/flox^ × *Lck-Cre* (G56KO) [21], *Zfp683*^−/−^ [22] or *Zfp683*^−/CRE^ (HKO) [23], *Zfp683^+^*^/CRE^ × *Prdm1*^flox/flox^ (BKO) mice [24], and *Zpf683*^−/CRE^ × *Prdm1*^flox/flox^ (DKO) were maintained on a C57BL/6JRj background under specific pathogen-free conditions in the animal facility of the Netherlands Cancer Institute (Amsterdam, The Netherlands). Animal experiments were conducted according to institutional and national guidelines.

### 2.2. Assessment of CRE Recombinase Activity at Adgrg1 Locus

CD8^+^ T cells were isolated from spleen by mouse CD8α MicroBeads (Miltenyi Biotec, Bergisch Gladbach, Germany). Cell lysates of toes and CD8^+^ T cells were obtained using lysis buffer (100 mM Tris-HCl, 5 mM EDTA pH 8.0, 0.2% (v/v) SDS, 200 mM NaCl, 200 μg/mL proteinase K (all reagents from Sigma-Aldrich, St. Louis, MO, USA)). Genomic DNA (gDNA) was isolated from cell lysates using isopropanol extraction. The following forward and reverse primers were used for amplification of *Adgrg1* locus (forward: 5′-GCAGATTCCCCAGAACACCA-3′, reverse: 5′-ACCCAAGACCTTCTCACCCA-3′). Amplification of gDNA was performed on a Veriti 96-Well Fast Thermal Cycler (Applied Biosystems, Waltham, MA, USA) using the DreamTaq Hot Start Green PCR Master Mix (Thermo Fisher Scientific, Waltham, MA, USA).

### 2.3. Lymphocytic Choriomeningitis Virus (LCMV) and Listeria Monocytogenes Bacterial Infection

Mice were infected intraperitoneally with 30 plaque-forming units (PFU) of the LCMV strain LCMV-WE or 1 × 10^5^ PFU of the LCMV strain LCMV-Armstrong or were infected by oral administration with 2.5 × 10^9^ colony-forming unit (CFU) of recombinant *Listeria monocytogenes* expressing OVA (*Listeria*-OVA) InlA^M^ (kindly provided by B. Sheridan, Stony Brook University). For rechallenge responses, mice that had been orally infected with 2.5 × 10^9^ *Listeria*-OVA InlA^M^ were reinfected 30 days later with a second dose of 2.5 × 10^10^ *Listeria*-OVA InlA^M^. At the indicated time points after infection, mice were sacrificed and organs were collected for analysis of CD8^+^ T-cell responses.

### 2.4. Tissue Preparation and Flow-Cytometric Analysis

Mononuclear cells from blood, spleen, liver, and gut, including small intestinal intraepithelial lymphocytes (SI-IELs) and lamina propria lymphocytes (LPLs), were isolated as described previously [25]. Cells were incubated with fluorochrome-conjugated antibodies and tetramers for 30 min at 4 °C and washed with PBS supplemented with 0.5% (v/v) fetal calf serum. Exclusion of dead cells was performed with live/dead fixable near-IR dead cell stain kit (Thermo Fisher Scientific, Waltham, MA, USA). Flow cytometry was performed using an LSRFortessa cell analyzer (BD Biosciences, San Jose, CA, USA) and FlowJo software (version 10; Tree Star, Ashland, OR, USA) using standard procedures and antibodies directed against CD3 (clone 17A2), CD4 (clone GK1.5 or RM4-5), CD8α (clone 53–6.7), CD8β (clone YTS156.7.7), CD62L (clone MEL-14), CD69 (clone H1.2F3), CD103 (clone M290), CD107a (clone 1D4B), CD127 (clone A7R34), granzyme B (clone GB-11), IFN-γ (clone XMG1.2), Ki-67 (clone 16A8), KLRG1 (clone 2F1), TCRβ (clone H57-597), and TCRγ/δ (clone UC7-13D5), purchased from BioLegend (San Diego, CA, USA), eBiosciences (San Diego, CA, USA), and BD Biosciences (San Jose, CA, USA). To detect LCMV-specific CD8^+^ T cells, MHC class I D^b^-restricted tetramers for the viral epitopes GP_33-41_ and NP_396−404_ were produced as described [26] (kindly provided by R. Arens, Leiden University Medical Center). To detect OVA-specific CD8^+^ T cells, MHC class I K^b^-restricted tetramers for the ovalbumin epitope OVA_257-264_ were used [27] (also kindly provided by R. Arens, Leiden University Medical Center). For the staining of intracellular molecules, the Foxp3/Transcription Factor Staining Buffer Set (eBioscience) was used according to the manufacturer’s specifications.

### 2.5. In Vitro CD8 T Cell Stimulation

Murine CD8^+^ T cells isolated from spleen and small intestine were activated in 96-well plates (Corning, Corning, NY, USA), coated with 10 μg/mL anti-CD3 (clone 145 2C11; BD Bioscience) and 2 μg/mL anti-CD28 (Clone 37.51; BD Bioscience), in the presence of 10 ng/mL IL-2, 10 ng/mL IL-7, and 10 ng/mL IL-15 with or without 10 ng/mL TGF-β (all from PeproTech, London, UK). After 3 days of culture, CD8^+^ T cells were replated and cultured for 6 additional days with only IL-2, IL-7, and IL-15. For the peptide stimulation assay, GP_33-41_-specific CD8^+^ T cells were activated in 96-wells plates in the presence of 5 μg/mL of the GP_33-41_ peptide KAVYNFATC for 5 hr. The anti-CD107a (clone 1D4B) antibody was added to assess degranulation. Brefeldin A and Monensin (both from eBioscience) were added to enable intracellular capture of IFN-γ and Cytofix Fixation Buffer (BD Bioscience) was used for staining of intracellular cytokines.

### 2.6. Quantitative RT-PCR

Total RNA was isolated with Trizol reagent (Thermo Fisher Scientific) or RNeasy Micro Kit (Qiagen, Hilden, Germany), and cDNA synthesis was performed on a Veriti 96-Well Fast Thermal Cycler (Applied Biosystems) using the iScript RT PCR kit (Bio-Rad, Hercules, CA, USA). Relative gene expression levels were measured via quantitative reverse transcription-polymerase chain reaction (RT-PCR) using Fast SYBR Green Master mix (Applied Biosystems) on a StepOnePlus system (Applied Biosystems) with the cycle threshold method. Primers for the following genes were used: *Adgrg1* (forward: 5′-CTGCGGCAGATGGTCTACTTC-3′, reverse: 5′-CCACACAAAGATGTGAGGCTC-3′) and *Hprt* (forward: 5′-TGAAGAGCTACTGTAATGATCAGTCAAC-3′, reverse: 5′-AGCAAGCTTGCAACCTTAACCA-3′). Expression values are represented relative to that of *Hprt* and calibrated relative to naive CD8^+^ T cells from spleen.

### 2.7. Statistical Analysis

All analysis was performed in GraphPad Prism 9 (GraphPad Software, San Diego, CA, USA) using one-way or two-way ANOVA test and Tukey’s multiple comparisons test.

## 3. Results

### 3.1. CD8^+^ T_RM_ Cells Specifically Upregulate Adgrg1/GPR56

To establish the expression profile of adhesion GPCRs in pathogen-specific T cells, we analyzed RNA sequencing data of memory CD8^+^ T cells that arose after herpes simplex virus (HSV) or lymphocytic choriomeningitis virus (LCMV) infection in mice. We observed that T cells only expressed 4 out of a panel of 31 adhesion GPCRs, including *Adgre5*/CD97 and the gene cluster *Adgrg1*/GPR56, *Adgrg3*/GPR97, and *Adgrg5*/GPR114. *Adgre5* and *Adgrg5* expression was upregulated in naïve CD8^+^ T cells and was retained in all analyzed memory CD8^+^ T-cell lineages (Figure 1A). In contrast, *Adgrg1* and *Adgrg3* tran-scripts were not expressed in naïve CD8^+^ T cells but were upregulated in virus-specific CD8^+^ T cells after HSV or LCMV infection (Figure 1A). Upregulation of *Adgrg3* was most pronounced in HSV-specific T_RM_ cells in skin, whereas expression of *Adgrg1* was strongly upregulated in HSV-specific T_RM_ cells of the skin and in LCMV-specific T_RM_ cells of the liver and the small intestine (Figure 1A). We did not find expression of *Adgrg1* in circulating memory CD8^+^ T cells, including T_CM_- and T_EM_-cell populations in the spleen and the liver (Figure 1A), indicating that this adhesion GPCR was specifically induced in pathogen-specific T_RM_ cells. The quantitative RT-PCR analysis validated the specific upregulation of *Adgrg1* in virus-specific T_RM_ cells after acute infection with LCMV (Figure 1B). The T_RM_ cell-specific upregulation of *Adgrg1* appeared to occur in the memory phase, given that we were unable to detect expression of *Adgrg1* in effector populations, including terminal effector cells (TECs) and memory precursor effector cells (MPECs) at day 8 after infection with LCMV (Figure 1B). Interestingly, T_RM_ cells also specifically express high levels of the transcription factor Hobit and the cytolytic enzyme granzyme B, in contrast to circulating T_CM_ and T_EM_ cells [25,28]. Therefore, *Adgrg1*/GPR56 expression appears to correlate with the expression of *Zfp683*/Hobit and the presence of granzyme B protein in murine as well as human CD8^+^ T cells.

### 3.2. CD8^+^ T_RM_ Cell-Inducing Factors Regulate Adgrg1/GPR56 Expression

We previously showed that Hobit and Blimp-1 strongly collaborate in the transcriptional regulation of T_RM_ cells [28] and that Hobit is the driving transcription factor for the expression of granzyme B in T_RM_ cells [25]. The analysis of *Adgrg1* expression in T_RM_ cells of *Zfp683*-deficient and *Prdm1*-deficient mice showed strong reduction of *Adgrg1* expression in mouse T_RM_ cells lacking Hobit and, to a lesser extent, in those lacking Blimp-1 (Figure 2A,B), in support of a role for Hobit in the transcriptional regulation of GPR56. The expression of *Adgrg1* was further reduced in *Zfp683* and *Prdm1* double-deficient T_RM_ cells (Figure 2B), indicating coregulation of Hobit and Blimp-1 in the maintenance of *Adgrg1* expression in T_RM_ cells. It is important to note that the reduced numbers of T_RM_ cells in Hobit and Blimp-1 double-deficient mice may also impact *Adgrg1* expression. However, the partial defects in GPR56 expression in single Hobit and Blimp-1 deficient mice that have normal numbers of T_RM_ in small intestine suggest a direct impact of these transcription factors on the regulation of GPR56 expression.

TGF-β has been shown to specifically promote the development or maintenance of T_RM_ cells in vivo [29] in the skin and small intestine [30]. TGF-β can also establish part of the residency-specific transcriptional profile of T_RM_ cells, including the induction of *Itgae*/CD103 [31]. To determine the impact of TGF-β on *Adgrg1* expression, we analyzed RNA sequencing data of spleen CD8^+^ T cells cultured in the presence of IL-2 and/or TGF-β. We observed that TGF-β not only induced expression of *Itgae* under these conditions but also strongly upregulated expression of *Adgrg1* (Figure 2C). The presence of IL-2 did not impact *Adgrg1* expression (Figure 2C). Analysis using the quantitative RT-PCR validated the upregulation of *Adgrg1* transcripts in splenic CD8^+^ T cells after culture with TGF-β (Figure 2D). Furthermore, the expression of *Adgrg1* is substantially reduced in skin T_RM_ cells of mice with a deficiency in the TGF-β receptor II (*Tgfbr2*^−/−^) (Figure 2E). Our results suggest that the T_RM_ cell-inducing transcription factor Hobit and the T_RM_ cell-inducing cytokine TGF-β contribute to the expression of *Adgrg1* in T_RM_ cells.

### 3.3. Adgrg1/GPR56 Is Dispensable for the Development of CD8^+^ T_RM_ Cells

To study the involvement of GPR56 in CD8^+^ T-cell differentiation, we crossed floxed *Adgrg1* mice [21] with *Lck-Cre* mice [24], thereby generating mice with deficient GPR56 expression in T cells. Analysis of the activity of *Lck*-driven CRE at the *Adgrg1* locus in T cells indicated the specific deletion of this gene (Appendix A). The impact of GPR56 was analyzed on virus-specific CD8^+^ T cells arising after an acute viral infection with LCMV (Figure 3A). Virus-specific CD8^+^ T cells recognizing the dominant epitope GP_33-41_ were detected using tetramers at different locations and time points after infection. We found that GPR56 was not essential for the formation of GP_33-41_^+^ CD8^+^ T cells in spleen, liver, and small intestine at effector (day 8) and memory time points (day 30+) after LCMV infection (Figure 3B). Expression of IL-7Rα chain (CD127) in conjunction with KLRG1 has been used to distinguish memory precursor effector cells (MPECs; KLRG1^−^CD127^+^) from terminal effector cells (TECs; KLRG1^+^CD127^−^) in mice [32,33]. Our analysis showed that these two effector populations appear to develop normally in the absence of GPR56 during the effector phase of an LCMV infection (Figure 3C). Expression of CD69 and CD62L divides memory CD8^+^ T cells into CD69^−^CD62L^+^ T_CM_, CD69^−^CD62L^−^ T_EM_, and CD69^+^CD62L^−^ T_RM_ cells. In contrast to T_RM_ cells in liver, a large fraction of T_RM_ cells in the intraepithelial compartment of the small intestine expressed CD103 [34]. Analysis of the memory subsets in WT and *Adgrg1*-deficient mice showed that the formation of T_CM_-, T_EM_-, and T_RM_-cell populations in the spleen and liver and the formation of CD103^−^ and CD103^+^ T_RM_ cells in the small intestine was not dependent on GPR56 (Figure 3D). Thus, GPR56 does not appear to impact the differentiation of virus-specific CD8^+^ T cells into memory precursors and the development of downstream T_CM_-, T_EM_-, and T_RM_-cell populations.

### 3.4. Adgrg1/GPR56 Does Not Regulate Cytokine Production and Release of CD8^+^ T_RM_ Cells

Virus-specific memory CD8^+^ T cells that develop after LCMV infection can effectively respond with the production and release of proinflammatory cytokines, such as IFN-γ. To examine whether GPR56 is involved in the regulation of IFN-γ production upon reactivation, we briefly cultured spleen and liver cells of LCMV-infected mice in the presence of the LCMV peptide GP_33-41_ (Figure 4A,B). Taking advantage of the T_RM_ cell-associated molecule CD69 to distinguish circulating memory T cells from T_RM_ cells, we observed that restimulated T_RM_ cells from liver produced more IFN-γ than circulating memory T cells from the spleen and liver (Figure 4A,B). Moreover, T_RM_ cells upregulated expression of CD107a, indicating the ability to degranulate and release cytokines, to a higher extent than circulating memory T cells (Figure 4A,B). However, IFN-γ production and degranulation was independent of GPR56 (Figure 4A,B). Taken together, we did not find evidence that GPR56 has an effect on the production and release of the proinflammatory cytokine IFN-γ by CD8^+^ T cells.

### 3.5. Adgrg1/GPR56 Does Not Regulate Cytotoxic Function of CD8^+^ T_RM_ Cells

We exploited the *Listeria*-ovalbumin (OVA) infection model to validate our findings on the role of GPR56 in T_RM_ cells in the LCMV infection model. *Listeria monocytogenes* is an intracellular bacterium that, upon oral administration, establishes acute infection in the small intestine and after systemic spread in other organs such as the liver. Similar as in the LCMV infection model, T_RM_-cell populations develop in liver and small intestine after *Listeria*-OVA infection [35]. We used tetramers recognizing OVA-specific CD8^+^ T cells to determine the effect of GPR56 on the differentiation of OVA-specific T cells after primary infection with *Listeria*-OVA (Figure 5A). Corroborating our results in the LCMV infection model, GPR56 did not appear to impact the differentiation of pathogen-specific CD8^+^ T cells into T_CM_, T_EM_, and T_RM_ cells in the spleen and liver or into CD103^−^ and CD103^+^ T_RM_-cell populations in the intraepithelial and lamina propria compartment of the small intestine after *Listeria*-OVA infection (Figure 5B). Effector CD8^+^ T cells upregulate protein expression of the cytolytic enzyme granzyme B, which is retained in T_RM_ cells after pathogen clearance, but not in circulating memory CD8^+^ T cells [25]. Consistent with these findings, we observed that granzyme B protein expression is elevated in T_RM_ cells in liver and small intestine compared to T_CM_ and T_EM_ cells in liver and spleen (Figure 5C). However, comparison between WT and *Adgrgr1*-deficient CD8^+^ T cells did not reveal differences in the expression of granzyme B (Figure 5C). Thus, GPR56 does not appear to essentially contribute to T_RM_-cell differentiation in the *Listeria*-OVA infection model and the regulation of granzyme B expression in *Listeria*-OVA specific T_RM_ cells.

### 3.6. Secondary CD8^+^ T Cell Responses Are Not Regulated by Adgrg1/GPR56

T_RM_ cells have the ability to expand and differentiate into effector cells that combat re-encountered pathogens upon reinfection. In fact, T_RM_ cells substantially contribute to the establishment of secondary CD8^+^ T-cell responses [23]. We therefore employed the *Listeria*-OVA infection model to study the role of GPR56 in T-cell responses in the context of prime boost infection (Figure 6A). We observed that secondary infection of *Listeria*-OVA resulted in a substantial increase in the percentage and number of pathogen-specific CD8^+^ T cells in the spleen, liver, and intraepithelial and lamina propria compartment of the small intestine (Figure 6B–E). This increase in the number of pathogen-specific CD8^+^ T cells was notable as early as day 8 after reinfection (Figure 6B–E). However, GPR56 did not appear to have an effect on the magnitude of the secondary CD8^+^ T cell response in spleen, liver, and small intestine (Figure 6B–E). We further analyzed the secondary response of pathogen-specific CD8^+^ T cells using expression analysis of the proliferation-associated marker Ki-67. In line with their re-expansion, we detected increased expression of Ki-67 in pathogen-specific CD8^+^ T cells at day 8 after reinfection with *Listeria*-OVA (Figure 6B–E). However, the expression of Ki-67 was not different between pathogen-specific CD8^+^ T cells of WT and *Adgrg1*-deficient mice (Figure 6B–E). In sum, GPR56 expression did not appear to have an effect on the proliferative responses of pathogen-specific CD8^+^ T cells upon secondary infection with *Listeria*-OVA. Thus, we conclude that GPR56, despite its highly specific expression on T_RM_ cells, does not essentially contribute to the regulation of the differentiation of T_RM_ cells after primary infection and their reactivation and re-expansion after secondary infection.

## 4. Discussion

In this report, we have studied the role of the inhibitory receptor GPR56 on pathogen-specific CD8^+^ T cells. We found that *Adgrg1* was specifically upregulated in CD8^+^ T_RM_ cells in the memory phase after infection with HSV or LCMV. Moreover, we established evidence that implicates the T_RM_ cell-associated transcriptional regulators Hobit and Blimp-1 and the T_RM_ cell-inducing cytokine TGF-β and in the upregulation of *Adgrg1* in CD8^+^ T_RM_ cells. However, under the setting of acute infection with LCMV and *Listeria monocytogenes*, we did not observe an essential inhibitory role of GPR56 in the regulation of proliferative, cytokine, or cytotoxic responses of CD8^+^ T_RM_ cells. We conclude that GPR56 is specifically upregulated on CD8^+^ T_RM_ cells but does not provide an essential contribution in the regulation of the proinflammatory activity of T_RM_ cells after acute infection.

Of note, the balance of GPR56 and GPR97 in hematopoietic stem cells (HSCs) is crucial for the development and differentiation of HSCs [36]. Therefore, it is possible that compensatory effects of other adhesion GPCRs such as GPR97 negate the impact of *Adgrg1*-deficiency on T_RM_ cells. In line with potential redundant functions between adhesion GPCRs in T_RM_, we found pronounced expression of *Adgrg3* in skin T_RM_ and ubiquitous expression of *Adgrg1* throughout the analyzed T_RM_-cell populations. The study of the combined role of these two adhesion GPCRs in T_RM_ cells is an important future research direction.

T_RM_ cells have been established as a separate lineage of memory CD8^+^ T cells with a unique transcriptional profile [33]. Here, we have identified GPR56 as a T_RM_ cell-associated receptor. *Adgrg1* was expressed in T_RM_ cells in skin, liver, and small intestine, but not in circulating T_CM_- and T_EM_-cell populations in spleen and liver. T_RM_-cell populations in different tissues are exposed to distinct environmental conditions and therefore may have tissue-specific expression profiles. The expression of *Adgrg1* in T_RM_ cells in skin, liver, and small intestine suggests that GPR56 is upregulated on T_RM_ cells in diverse microenvironments and may be part of the universal gene signature of T_RM_ cells. However, further analysis of T_RM_ cells in other tissues is required to determine whether GPR56 is ubiquitously expressed in T_RM_-cell populations throughout tissues.

In line with conserved expression of GPR56 between T_RM_-cell populations of humans and mice, we recently reported GPR56 expression on human brain T_RM_ cells [37]. Although GPR56 has a wider expression pattern in humans than in mice, the retained expression of GPR56 in human T_RM_-cell populations suggests that our findings in mice have relevance for humans. Importantly, we identified expression of GPR56 protein in human brain T_RM_ cells, indicating that this adhesion GPCR, at least in humans, is not regulated at protein level [37]. Unfortunately, antibodies are not yet available for murine GPR56, preventing us from analyzing protein expression in mice. Therefore, the possibility that lack of protein expression underlies the absence of functional defects in T_RM_ cells of GPR56-deficient mice exists. However, we consider this option unlikely, given that we can detect both GPR56 mRNA and protein in humans.

The expression of *Adgrg1* appears to be induced in the T_RM_-cell lineage at a late time point during T-cell differentiation after acute infection. We did not observe expression of *Adgrg1* in naïve T cells or in effector CD8^+^ T cells including memory precursors that are upstream of T_RM_ cells. We cannot exclude the possibility that a minor subset of these memory precursors upregulates *Adgrg1* expression. Memory precursors with exclusive potential to form T_RM_ cells appear to separate early from other effector cells [38,39,40]. These T_RM_ precursor cells may be present in the bloodstream [40] and locally within peripheral tissues [38], where they eventually settle and form T_RM_ cells. With current tools, it remains difficult to unequivocally identify precursor stages of T_RM_ cells; therefore, analysis of the expression of GPR56 by these subsets awaits further characterization of T_RM_ precursor cells. The repertoire of surface markers to identify T_RM_ cells includes CD69, CD103, CD49a, and P2XR7 and other surface molecules [33,41,42]. Our identification of the specific expression of GPR56 on T_RM_ cells suggests that this surface receptor may also be employed as a surrogate marker of T_RM_ cells upon the establishment of antibodies recognizing mouse GPR56. As mentioned above, the expression pattern of GPR56 in humans appears broader than in mice. In humans, GPR56 is also expressed on cytotoxic lymphocytes, including NK cells and effector-type CD8^+^ and CD4^+^ T cells in peripheral blood [4] besides T_RM_ cells in peripheral tissues [37]. The specific expression of GPR56 on T_RM_ cells, rather than circulating memory T cells, in mice is consistent with their immediate cytotoxic capacity, given that T_RM_ cells retain cytotoxic molecules, such as granzymes, at protein level in contrast to other memory T cells [25]. Thus, although GPR56 in humans may be differentially expressed compared to GPR56 in mice, upregulation of this adhesion GPCR on the cytotoxic fraction of memory T cells appears conserved between both species.

The differentiation of T_RM_ cells is instructed by cytokines in their local environment. In particular, TGF-β has been found instrumental in the differentiation of T_RM_ cells in the epithelial compartment of the skin and the small intestine [29,33]. We have identified TGF-β as a potent inducer of *Adgrg1* expression in CD8^+^ T cells in in vitro cultures. Notably, developing T_RM_ cells in skin require TGF-β signaling to acquire *Adgrg1* expression. These findings suggest that T_RM_-cell populations in the skin and small intestine upregulate GPR56 expression in response to TGF-β signaling. Expression of CD103, the αE component of the αEβ7 integrin, also strongly depends on TGF-β signaling [29]. In contrast to CD103, we have observed that GPR56 is also expressed on liver T_RM_ cells outside of CD103-expressing T_RM_-cell populations in skin and intestine. Therefore, the expression regulation of GPR56 in T_RM_ cells appears more complex and may include environmental cues other than TGF-β. We have previously observed in human NK cells that the expression of GPR56 is downregulated in the presence of the homeostatic cytokine IL-15 and upon activation with inflammatory cytokines, including IL-2 and IL-18 [10]. Therefore, it is unlikely that IL-15, which can contribute to T_RM_-cell maintenance in mice [43,44,45], and these other cytokines play a role in the upregulation of GPR56 on T_RM_ cells. We have previously also observed that GPR56 was induced by the transcription factor Hobit in human NK cells [10]. Hobit and the related transcription factor Blimp-1 are master regulators of T_RM_-cell differentiation, which, through suppression of tissue exit molecules, such as S1PR1 and CCR7, can permanently lock these memory T cells into the peripheral tissues [28]. Consistent with these findings, we observed here that Hobit together with Blimp-1 contributed to the transcriptional regulation of GPR56 expression in T_RM_ cells. Hobit and Blimp-1 are broadly expressed in T_RM_ cells throughout tissues including those of the skin, liver, and small intestine [28], suggesting that the transcription factor may drive GPR56 expression on CD103^−^ and CD103^+^ T_RM_-cell populations in these tissues. The expression of Hobit and Blimp-1 appears independent of TGF-β signaling, but currently, it remains unknown how the expression of these transcription factors is upregulated during T_RM_-cell differentiation. Thus, GPR56 expression may be regulated through separate pathways, involving TGF-β and yet unresolved cytokines that trigger Hobit and Blimp-1 expression.

The strong proinflammatory activities of T_RM_ cells after reinfection may have harmful impact on the surrounding healthy tissues. Pathogen-specific T_RM_ cells express a multitude of inhibitory receptors including PD-1, TIM-3, LAG-3 and CD39 that can restrain T_RM_ cell-driven immune responses [33]. Interestingly, CD103^+^ T_RM_ cells appear to be under stronger regulation by inhibitory receptors than CD103^−^ T_RM_ cells [30]. In line with these findings, we also found that the inhibitory receptor GPR56 is responsive to TGF-β suggesting elevated expression of this receptor on CD103^+^ T_RM_ cells compared to CD103^−^ T_RM_ cells. We have previously found that GPR56 is an inhibitory receptor, which can suppress the cytotoxic and cytokine responses of NK cells [10]. In this paper, we addressed the suppressive function of GPR56 on T_RM_ cells after infection with LCMV and *Listeria monocytogenes*. In contrast to its essential role in regulating proinflammatory responses of NK cells, we did not observe that GPR56 substantially contributed to the control of the magnitude of T_RM_-cell responses or the cytokine production of T_RM_ cells. We also did not find evidence that GPR56 contributed to the regulation of the cytolytic enzyme granzyme B in T_RM_ cells, suggesting that the regulatory role of GPR56 is not essential under these conditions. Evidence is accumulating that expression of inhibitory receptors, such as PD-1, on T_RM_ cells is highly relevant to protect against T_RM_ cell-driven inflammatory responses that otherwise may develop in the intestine, pancreas, and the lungs [18,19,20]. Possibly, the inhibitory impact of GPR56 in the regulation of T_RM_ cells is masked by the presence of other inhibitory receptors on T_RM_ cells. Given that proinflammatory cytokines suppress GPR56 expression [10], unleashing the inhibitory potential of GPR56 may also require an anti-inflammatory environment, such as occurs in a tumor setting. More recently, T_RM_ cells have been detected in tumor tissue of mouse models of melanoma [46], and in patients with melanoma, lung carcinoma, and ovarian carcinoma [47,48,49]. These tumor-residing T_RM_ cells highly express PD-1, and inhibition of the PD-1-driven checkpoint blockade pathway appears to reinvigorate their antitumor activity [49]. Therefore, inhibitory receptors appear relevant on T_RM_ cells to restrain antitumor immune responses. Our findings show that T_RM_ cells also express dedicated inhibitory receptors such as GPR56 that suggest the presence of unique regulatory mechanisms in T_RM_ compared to circulating memory T cells. However, resolving the potential role of GPR56 on tumor-resident T_RM_ cells awaits future research.

## Figures and Tables

**Figure 1 cells-10-02675-f001:**
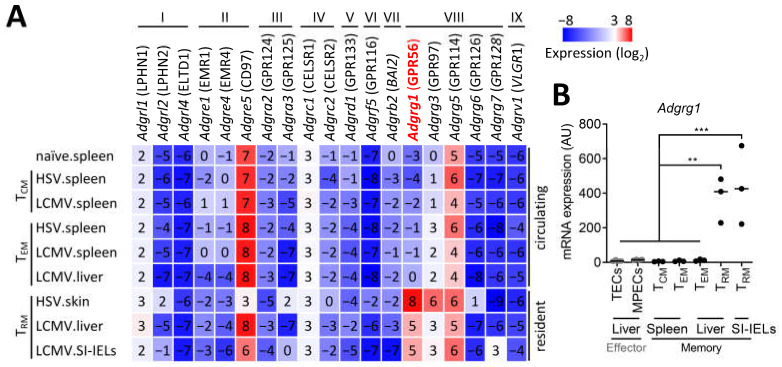
CD8^+^ TRM cells express *Adgrg1*/GPR56. (**A**) Naïve CD8^+^ T cells were sorted from uninfected mice, and virus-specific memory CD8^+^ T-cell populations were sorted from indicated tissues of mice, infected with herpes simplex virus (HSV) or lymphocytic choriomeningitis virus (LCMV), at day 40+ postinfection (memory phase). A panel of 31 adhesion GPCRs was analyzed, of which 19 adhesion GPCRs were present, and 12 adhesion GPCRs were absent in the RNA sequencing data. Data are derived from GSE70813. (**B**) Adgrg1 transcripts were determined by quantitative RT-PCR in indicated CD8^+^ T-cell populations from liver, spleen, and small intestinal intraepithelial lymphocytes (SI-IELs). LCMV-specific CD8^+^ T cells were collected at different time points of infection. Terminal effector cells (TECs) and memory precursor cells (MPECs) were isolated at day 8 (effector phase), and memory CD8^+^ T cells were obtained at day 30+ postinfection (memory phase). The sorting strategy is shown in Appendix A. One-way ANOVA with Tukey’s multiple comparisons test; ** *p* < 0.01, *** *p* < 0.005.

**Figure 2 cells-10-02675-f002:**
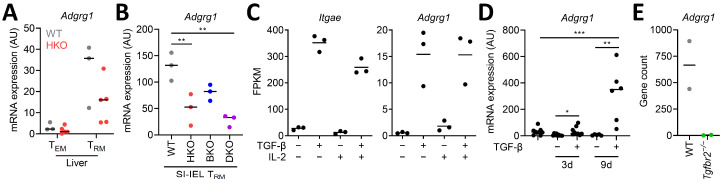
*Adgrg1*/GPR56 expression in CD8^+^ T_RM_ cells requires Hobit and TGF-β. (**A**) *Adgrg1* transcripts were determined using quantitative RT-PCR in LCMV-specific CD8^+^ T_EM_ and T_RM_ cells from liver of wild type (WT, ●) and *Zfp683*-deficient (HKO, ●) mice at day 40+ after LCMV-WE infection. The gating strategy is shown in Appendix A. (**B**) *Adgrg1* transcripts were determined using quantitative RT-PCR in LCMV-specific CD8^+^ T_RM_ cells from small intestine intraepithelial lymphocytes (SI-IELs) of WT, *Zfp683*^−/CRE^ (HKO), *Zfp683*^+/CRE^ × *Prdm1*^flox/flox^ (BKO), and *Zfp683*^−/CRE^ × *Prdm1*^flox/flox^ (DKO) mice at day 30+ after LCMV-Armstrong infection. The sorting strategy is similar, as shown in Appendix A. (**C**) *Itgae* and *Adgrg1* transcript counts are shown upon different condition treatments as indicated. Data are derived from GSE125471. FPKM, fragments per kilobase of transcript per million mapped reads. (**D**) *Adgrg1* transcripts were determined using quantitative RT-PCR in sorted spleen CD8^+^ T cells after culture with or without TGF-β for 3 and 9 days from two independent experiments (*n* = 6). (**E**) *Adgrg1* transcript counts are shown for OTI cells from WT or *Tgfbr2*-deficient mice in skin of HSV-OVA-infected recipients. Data are derived from GSE178769. One-way or two-way ANOVA with Tukey’s multiple comparisons test; * *p* < 0.05, ** *p* < 0.01, *** *p* < 0.005.

**Figure 3 cells-10-02675-f003:**
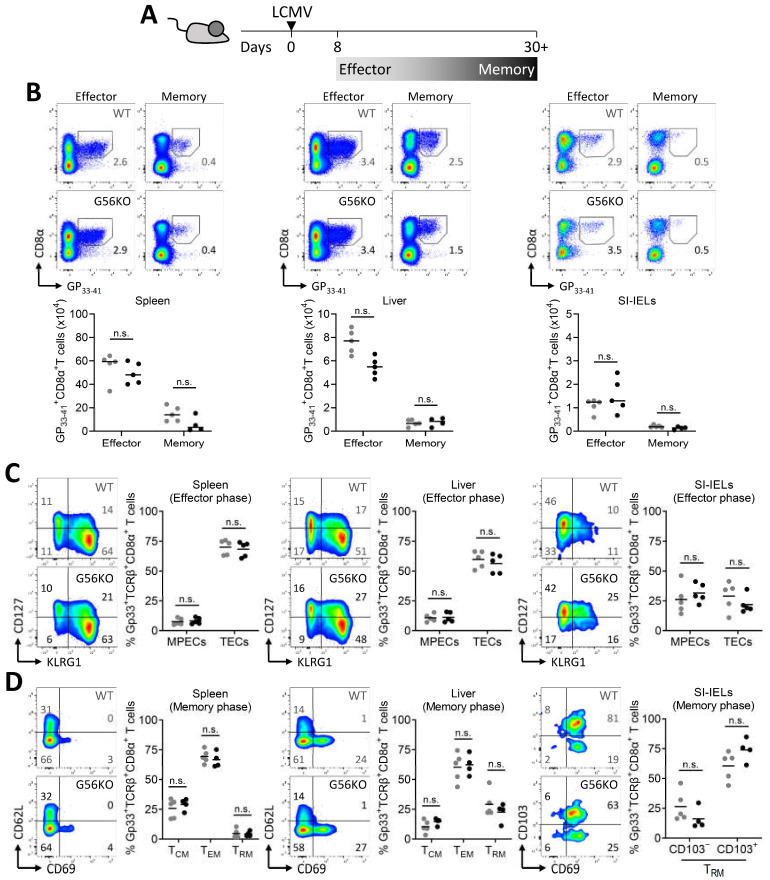
*Adgrg1*/GPR56 is dispensable for the formation of virus-specific CD8^+^ T_RM_ cells. (**A**) Experimental scheme shows the generation of LCMV-specific CD8^+^ T cells and indicates the effector and memory time points for analysis of virus-specific CD8^+^ T cells after acute LCMV-Armstrong infection. (**B**) Representative plots and absolute counts of virus-specific (GP_33-41_^+^) TCRβ^+^CD8^+^ T cells in wild-type (WT, ●) and *Adgrg1*-deficient (G56KO, ●) spleen, liver, and small intestinal intraepithelial lymphocytes (SI-IELs) at day 8 (effector phase) and 30+ (memory phase) after LCMV-Armstrong infection. (**C**) Representative flow cytometry plots and frequencies of virus-specific GP_33-41_^+^TCRβ^+^CD8^+^ memory precursor effector cells (MPECs, KLRG1^−^CD127^+^) and terminal effector cells (TECs, KLRG1^+^CD127^−^) in WT (●) and G56KO (●) spleen, liver, and SI-IELs at day 8 post-infection. (**D**) Representative flow cytometry plots and frequencies of virus-specific GP_33-41_^+^TCRβ^+^CD8^+^ CD69^−^CD62L^+^ T_CM_, CD69^−^CD62L^−^ T_EM_, CD69^+^CD62L^−^ T_RM_, and CD103^−/+^CD69^+^ T_RM_ cells in WT (●) and G56KO (●) spleen, liver, and SI-IELs at day 30+ post-infection. Two-way ANOVA with Tukey’s multiple comparisons test; n.s., no significant difference.

**Figure 4 cells-10-02675-f004:**
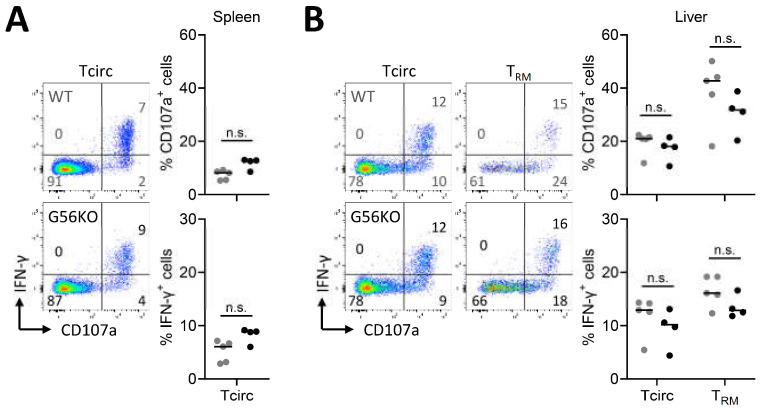
Adgrg1/GPR56 is dispensable for the function of virus-specific CD8^+^ T_RM_ cells upon in vitro stimulation with peptide antigen. (A, B) Representative flow cytometry plots and graphs displaying CD107a and IFN-γ expression of CD8^+^ CD69^−^ circulating T (Tcirc) and CD69^+^ T_RM_ cells in wild-type (WT, ●) and Adgrg1-deficient (G56KO, ●) spleen (**A**) and liver (**B**) upon in vitro restimulation with GP_33-41_ peptide. Two-way ANOVA with Tukey’s multiple comparisons test; n.s., no significant difference.

**Figure 5 cells-10-02675-f005:**
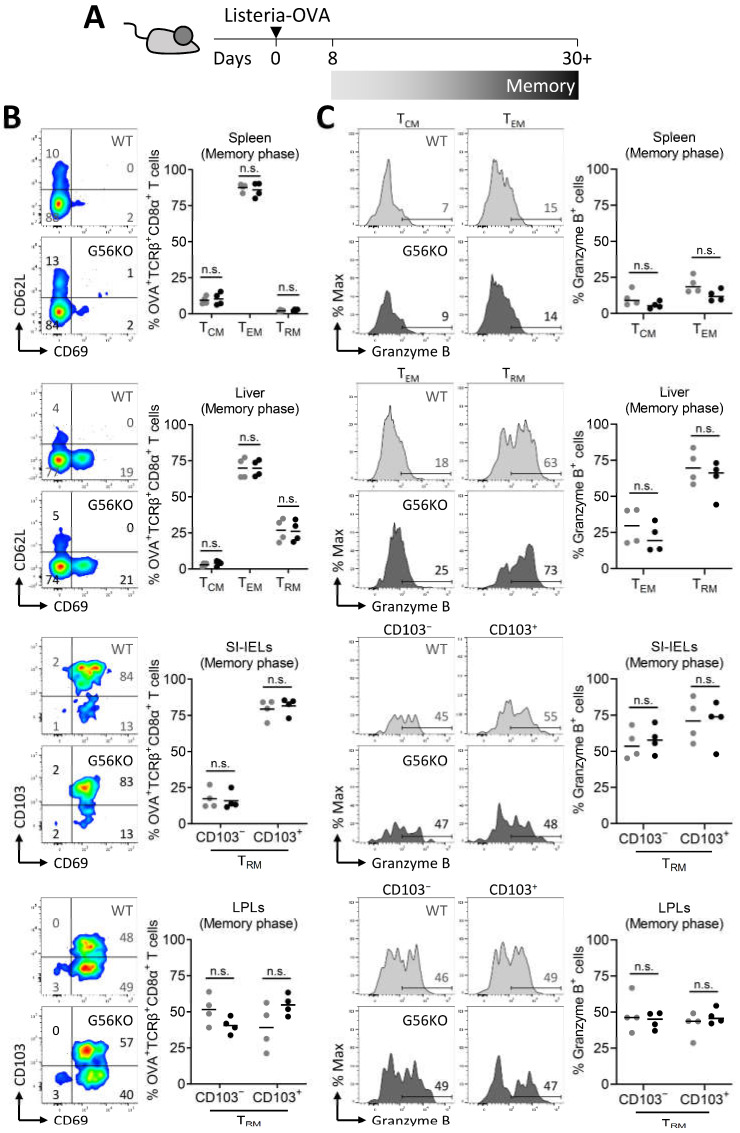
*Adgrg1*/GPR56 is dispensable for the formation and function of *Listeria*-specific CD8+ T_RM_ cells upon rechallenge. (**A**) Experimental scheme shows the generation of *Listeria*-OVA-specific memory T cells. (**B**) Representative flow cytometry plots and frequencies of *Listeria*-specific (OVA+) TCRβ^+^CD8^+^ CD69^−^CD62L^+^ T_CM_, CD69^−^CD62L^−^ T_EM_, CD69^+^CD62L^−^ T_RM_, and CD103^−/+^ CD69^+^ T_RM_ cells in wild-type (WT, ●) and *Adgrg1*-deficient (G56KO, ●) spleen, liver, and small intestinal intraepithelial lymphocytes (SI-IELs) at day 30+ postinfection. (**C**) Representative histograms and frequencies of granzyme B in indicated subsets of *Listeria*-specific (OVA^+^) TCRβ^+^CD8^+^ T_CM_, T_EM_, and T_RM_ cells in WT (●) and G56KO (●) spleen, liver, SI-IELs, and lamina propria lymphocytes (LPLs). Two-way ANOVA with Tukey’s multiple comparisons test; n.s., no significant difference.

**Figure 6 cells-10-02675-f006:**
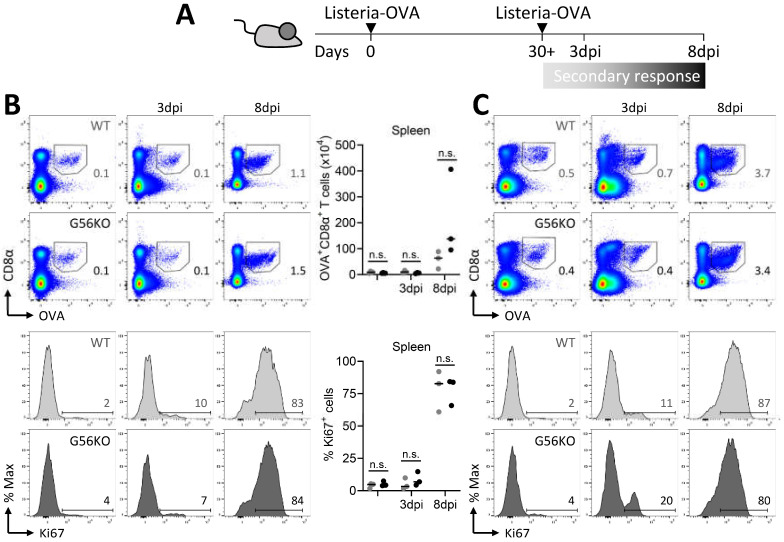
*Adgrg1*/GPR56 is dispensable for the expansion of *Listeria*-specific CD8^+^ T_RM_ cells upon rechallenge. (**A**) Experimental scheme shows the time points of analysis of *Listeria*-OVA-specific T cells after primary challenge and after-challenge with *Listeria*-OVA infection. (**B**–**E**) Representative flow cytometry plots and absolute counts of *Listeria*-specific (OVA^+^) CD8^+^ T cells and representative histograms of Ki-67 expression in *Listeria*-specific (OVA^+^) CD8^+^ T cells in wild-type (WT, ●) and *Adgrg1*-deficient (G56KO, ●) spleen (**B**), liver (**C**), small intestinal intraepithelial lymphocytes (SI-IELs) (**D**), and lamina propria lymphocytes (LPLs) (**E**) ex vivo and upon 3 and 8 days after rechallenge infection (3 and 8 dpi). Two-way ANOVA with Tukey’s multiple comparisons test; n.s., no significant difference.

## Data Availability

The data that support the findings of this study are available from the corresponding author upon reasonable request.

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
