# Peer review of "The Inhibitory Receptor GPR56 (Adgrg1) Is Specifically Expressed by Tissue-Resident Memory T Cells in Mice But Dispensable for Their Differentiation and Function In Vivo"

_cells, 2021, doi:10.3390/cells10102675_

Round 1

Reviewer 1 Report

I have reviewed the revised version of this manuscript and the authors have answered my concerns. 

Reviewer 2 Report

The Authors have performed significant amount of corrections and improved the overall discussion part. The manuscript looks fit for the publication.

Reviewer 3 Report

I am glad to see that this manuscript have been improved. The authors have properly addressed my previous comments and questions. I am convinced that the paper is suitable for publication in this form, therefore I support its appearance in my best opinion.

This manuscript is a resubmission of an earlier submission. The following is a list of the peer review reports and author responses from that submission.

Round 1

Reviewer 1 Report

    In this manuscript, Hsiao et al. observed that adhesion G protein coupled receptor GPR56 is specifically expressed and up-regulated in mouse tissue-resident memory T cells (TRM) upon reinfection. They also investigated its potential role in TRM differentiation and functions such as cytokine production and cytotoxic activity using GPR56 knock out mice. However, they were not able to show appreciable differences between wild type and GPR56 knock out mice, indicating that GPR56 is unlikely to play an essential role in TRM differentiation and functions, at least under pro-inflammatory conditions tested in this study.

    There are a few concerns with this manuscript:

  1. Overall, the negative results on TRM differentiation and function makes GPR56 role in this cell population inconclusive, which severely limits the significance of this study. Moreover, as pointed out by the authors in the discussion, the expression profiles of GPR56 in T cells are different between mouse and human. This further restrains the relevance of this research to human T cell biology and diseases.
  2. Though it is understandable that mouse GRP56 antibody might not be available to directly assess protein expression (especially surface expression) of GPR56 in TRM cells. Nevertheless, mRNA up-regulation does not always lead to protein up-regulation. The reason why the authors did not see functional impact of GPR56 in TRM could be that there are other layers of regulation in protein level that is not revealed in this study.
  3. Upon closer inspection, in some figures the difference between WT and G56KO seemed quite drastic (e.g. Figure 5B LPLs memory phase, Figure 6B spleen, though less evident). The statistical significance was not explicitly marked on the figures except Figure 2C. Even if the results are not significant, it should be marked as “ns” to avoid confusion.
  4. As for future investigation, it is interesting to see if inhibiting other inhibitory receptors on TRM (if inhibitor available) can unmask GPR56 functions. It could also be that GPR56 is functionally redundant. Since the authors also observed up-regulation of ADGRG3/GRP97 in some of the TRM populations, a double KO might reveal some functional defects.

Reviewer 2 Report

Minor comments:

  1. Line 113, the sentence needs to be revised.
  2. Line 181, the sentence needs to be revised.
  3. In Figure1 and 2, the RNAseq data from previous study need access number.
  4. In Figure 6, dose dpi stand for days post re-challenge? the author needs proper abbreviation.
  5. All the data in the figures need statistical analysis, even there are no significance, it need to be marked.

Reviewer 3 Report

The manuscript entitled “The inhibitory receptor GPR56 (Adgrg1) is specifically expressed……” is submitted for the publication in “Cells” by Hsiao et al. In this paper, authors have shown that TRM cells specifically acquire the inhibitory receptor GPR56 but the impact of this receptor on TRM cells after acute infection does not appear essential to regulate effector functions of TRM cells.

Overall, the findings of the paper are interesting and of high importance in context of Tissue residential memory cell biology in context to infection and immunity. The suppressive function of GPR56 on TRM cells is very well shown in the infection model with applicable dataset and appropriate experiments. I agree to the point that the manuscript should be accepted with some minor modification/correction.

Specifically:

- Kindly add all the frequency digits on the respective FACS plots in ALL the figures.

- In the Figure. 1 please add the plots for MPEC and SLEC which was used for isolation and subsequently for mRNA expression analysis

Reviewer 4 Report

In this manuscript, the authors explore the expression and function of inhibitory receptor GPR56 in the novel population of CD8+ TRM cells. Using HSV and LCMV infection GPR56 deficient mouse models, the authors report that GPR56 expresses in TRM cells upon acute infection, and the expression patten could be upregulated by the TRM cell-inducing cytokine TGF-β and the TRM cell-associated transcription factor Hobit. Interestingly, the expression of GPR56 has no effect to the regulation of proliferative, cytokine and cytotoxic responses of CD8+ TRM cells. Overall, this study shows GPR56 specifically required for the TRM cells but is not essential for the regulation effects upon acute infection.

A few concerns should be addressed:

Line 92: “Listeria immune mice received at day 30+ post infection a second oral dose of 2.5 x 1010 CFU Listeria-OVA InlAM”. This sentence needs to be rephrased.

Line 143: “Figure 1A” should be bold.

Line 150: “terminal effector cells (TECs)” should be consistent with the Figure 1B legend.

Line 170: “The remaining expression of Adgrg1 in Znf683-deficient TRM cells indicates the involvement of additional transcription factors, such as Blimp-1.” The statement should be supported by experiments, like the Hobit results shown in Figure 2A. It is interesting whether the expression of Adgrg1 would be downregulated upon Blimp-1 deficient.

Figure 2C: the repeats of the experiments n should be indicated.